

# Effects of restoration years on soil nitrogen and phosphorus in inland salt marshes

Dandan Zhao, Daiji Wan, Jian Yang, Jiping Liu, Zhicheng Yong and Chongya Ma

School of Geographical Sciences and Tourism, Jilin Normal University, Siping, Jilin, China

## ABSTRACT

Inland salt marsh wetlands have very important ecological functions in semi-arid areas. However, degradation and soil desertification have impacted these areas, making it necessary to study the impact of wetland restoration years on the soil quality of salt marsh wetland. We used remote sensing methods, field surveys, and inquiries to examine the seasonal profile effects of two-, four-, and six-year restoration periods on total nitrogen (TN), total phosphorus (TP) and the ratio of nitrogen to phosphorus (N:P) in *P. australis* and *S. triqueter* wetland natural states. Our results showed that soil TN in *P. australis* wetland in restored conditions was higher than that in natural conditions. The average soil TP of the *S. triqueter* wetlands at 0–10 cm, 10–20 cm, 20–30 cm, and 30–40 cm layers was 0.36 g/kg, 0.31 g/kg, 0.21 g/kg, and 0.17 g/kg s in September, respectively. The soil TP of the *S. triqueter* wetland increased slightly over the entire growing season. The restoration years had a great influence on the soil TP of the *S. triqueter* wetland from May to July. The soil TN in the *P. australis* wetland was almost restored to its natural condition in each layer during the six-year restoration period. The soil TP of the *S. triqueter* wetland was higher in the restored two-year period and showed a decreasing trend with an increased soil depth. Our conclusions can significantly guide the restoration of inland salt marsh wetlands.

## INTRODUCTION

Inland salt marsh wetlands are rich in salinity with alkaline surface water, and these special ecological conditions give rise to unique landscapes and ecosystems (*Bai et al., 2017*; *Cahill et al., 2021*). Recently, climate change and human activities have affected the desertification and salinization of these ecosystems, leading to the shrinkage of salt marsh wetlands (*Li, Li & Liu, 2020*). The circulation pattern of nutrients such as nitrogen and phosphorus has changed in wetland ecosystems in response to these activities. The natural ecological balance has been disturbed, thus affecting and destroying the stability of wetland ecosystems, resulting in the continuous decline of wetland productivity (*Liu et al., 2021*; *Pan et al., 2021*). The restoration of salt marsh wetlands greatly improves the structure and function of the above-ground and under-ground parts of the ecosystem, and has a

Corresponding author
Jiping Liu, liujpjl@163.com

protective effect on the biodiversity and sustainable development of the environment (*Xiong et al., 2022*; *Zhang et al., 2021*). The environmental characteristics and soil nutrient content improves when the soil is restored compared with degraded wetlands (*Jin et al., 2021*). The nutrient content of the soil has been shown to significantly affect the productivity of wetland ecosystems (*Zhang et al., 2022*). The nitrogen and phosphorus contents are crucial to wetland functions (*Walton et al., 2020*). Thus, they should be considered as important indexes in the evaluation of the success of wetland restoration (*Shan et al., 2018*). The organic matter content of the wetland surface soil increases linearly in the restoration years, and it takes more than 20 years to reach the level of a natural wetland (*Meyer & Whiles, 2008*). It is clear that wetland restoration is a long process; therefore, determining the mechanisms by which restoration years affect salt marsh wetland soils is a key priority.

Previous studies have found that wetland restoration restored nutrient elements to a certain extent and made up for the loss of soil nutrients caused by degradation (*Xu et al., 2021*). Carbon (C), nitrogen (N) and phosphorus (P) are the basis of the material composition of all living organisms on earth, and are reflected in the elemental composition of the living organisms in a functioning ecosystem (*Wang et al., 2021*). Their mutual coupling relationship is of great significance for maintaining the structure and functional stability of an ecosystem (*Dibar et al., 2020*). Soil N/P is an important index for predicting soil nutrient restriction and soil N saturation (*Zhao et al., 2015*). Wetland soils have izmportant ecological functions and are considered to be the source, sink and converter of N and P (*Simmons, 2018*). They play an important role in the balance and maintenance of wetland ecosystems and influence carbon sink, water storage, and plant diversity (*Shan et al., 2018*). After wetland restoration, the soil nutrient content improved compared with degraded wetlands, however, the land was not fully restored to its natural state (*Suir et al., 2019*). Many studies on the restoration of soil C have been conducted in wetland restoration (*Liu, Shi & Liang, 2019*; *Zhao et al., 2017*); however, the status of N and P in restored soil in salt marsh wetlands requires further study.

The soil N and P contents change seasonally, and the distribution characteristics of N and P reflect the nutrient supply, which, in turn, influences the composition of the plant community and the stability of wetland ecosystem (*Liu, Shi & Liang, 2019*). *Bai et al. (2007)* studied the seasonal dynamic characteristics of N and P in seaway wetlands and found that the contents of total nitrogen (TN) and total phosphorus (TP) showed a downward trend in the peak season of vegetation growth and accumulated in the mature and dead seasons. The content of TP in wetland soil showed a power function increase with the increase of organic matter content (*Bai et al., 2011*). Plants require more N and P during the peak growing season, which causes the contents of N and P in the soil to decrease. N and P increase in the soil during the mature and dormant seasons when plants stop absorbing these nutrients and instead transfer their above-ground nutrients into the soil (*Li et al., 2018a*). *Kong et al. (2014)* studied the restoration of converted farmland wetlands with different restoration periods along the Yangtze River in China and found that P needed a longer restoration time than N. The changes of soil nutrient contents and dynamics significantly affected the productivity of the wetland ecosystem; however, the

effects of restoration years on the seasonal changes of soil N and P in salt marsh wetlands remain unclear.

The N and P contents in wetland soil change significantly with soil depth (*Li et al., 2018a*) and vary due to climate, hydrological conditions, vegetation types, and the intensity human disturbance in different study areas (*Liu, Rong & Zhao, 2017*; *Msofe et al., 2019*). Studies have shown that although there are differences in N and P contents in wetland soil, their distribution in soil profile shows a consistent rule (*Wu et al., 2020*). Deeper soils reveal an increased TN content in marsh soil and a decreasing TP content. In the Zoige Natural Wetland Reserve of China (*Ye et al., 2016*), TN and TP in *larch-Carex*, *sedge*, and *Cyperus sinensis* wetlands increased first and then decreased with the increase of soil depth (0–40 cm) in the mountainous area of eastern Jilin Province, China (*Xiao et al., 2014*). The TN content of wetland soils first decreased and then increased with deeper soils, while TP gradually decreased with increased soil profile depths in the Yellow River Delta of China (*Li, Shi & Shao, 2018b*). However, the effects of restoration years on the vertical structure of TN and TP in salt marsh wetlands need to be further studied.

The western Songnen Plain is a concentrated distribution area of inland salt marsh wetlands; it is a semi-humid and semi-arid transition area that is important in regulating climate, storing water, and maintaining a regional ecological balance (*Qi et al., 2022*). Wetland productivity and biodiversity have declined continuously in recent years due to serious salinization and desertification (*David et al., 2016*; *Zhao & Liu, 2022*). We studied the typical plant communities of the Chinese wetlands (*P. australis* and *S. triqueter*) in the Xianghai National Nature Reserve in the western Songnen Plain. We took regular field samples and tests to determine the TN and TP contents of typical marsh wetlands in natural conditions, and restored two-, four-, and six-year periods. In our study, we tested whether (1) different restoration periods had seasonal variations in TN and TP in the soil; (2) whether there were structural differences in the vertical depths of TN and TP in wetland soil within different restoration periods; (3) effects of different restoration periods on the N:P of different wetlands. We analyzed the seasonal and structural characteristics of TN and TP over different restoration years and explored the influence of different restoration years on soil biogenic elements of inland salt marsh wetlands. Our results are of great significance and reveal the productivity, stability of the wetland ecosystem, and the restoration effect of marsh wetland.

## MATERIALS & METHODS

### Study area

The Xianghai National Nature Reserve in Songnen Plain is located in Tongyu County in the west of Jilin Province of China with a total area of 105,467 hm$^2$; its location is 122°05′–122°35′E, 44°50′–45°1′N. The landform type is mainly plain and located in the semi-arid region of eastern China, with typical continental monsoon climate characteristics. The average annual evaporation is 900–1,100 mm, and the average annual precipitation is 370–450 mm. The evaporation is far higher than the precipitation and the region is semi-arid (*Ding et al., 2017*). The soil thickness is generally between 0.5–1.0 m, and the

**Table 1** Geographic coordinates of soil sampling centers.

| Marsh wetland | | Geographic coordinates of sampling centers |
|---|---|---|
| *Phragmites australis* | Natural condition | 45°0′21.530′N, 122°20′12.811′E |
| | Restored two years | 45°0′11.259′N, 122°20′21.580′E |
| | Restored four years | 45°0′15.940′N, 122°20′36.403′E |
| | Restored six years | 45°0′18.129′N, 122°20′3.342′E |
| *Scirpus triqueter L.* | Natural condition | 45°1′53.520′N, 122°15′6.747′E |
| | Restored two years | 45°1′50.990′N, 122°15′39.854′E |
| | Restored four years | 45°1′51.644′N, 122°15′25.384′E |
| | Restored six years | 45°1′53.420′N, 122°15′4.206′E |

pH value is between 7.5–8.5. In the southern part of the reserve, there are large areas of saline-alkali land with small amounts of saline-alkali land, mostly found around marshes and lakes (*Liu, Li & Zhang, 2018*). *P. australis* and *S. triqueter* dominate the marsh wetland plants in this ecosystem. *P. australis* begins to green in May each year and enters its rapid growth period from May to June.

## Sampling plots

July to August is the growth season; plants begin to undergo apoptosis and gradually enter the wilting period from September to October. *P. australis* is a perennial marsh plant with well-developed roots, while Scirpus scirpus is a wet plant that is straight and upright and often found in furrows or marshes. Wetland restoration was accomplished by planting wetland plants in experimental plots. We selected *P. australis* and *S. triqueter* as two typical wetland plant communities for sampling and research. We utilized remote sensing, field surveys, and inquiries from Xianghai National Nature Reserve to confirm the extent of *P. australis* and *S. triqueter* in their natural state and at restored periods of two-, four-, and six-years. We selected the sampling points of restored wetlands to maintain the same natural wetland condition, such as geomorphic position, water level, human disturbance and other factors. We studied the ecological biogenic characteristics of typical plants (*P. australis* and *S. triqueter*) in restored and natural wetlands in Xianghai National Nature Reserve in different restoration years (two years, four years, six years). The specific geographical coordinates of plant sampling centers are shown in Table 1.

## Soil sample collection and processing

We used an undisturbed soil sampler with a liner to take soil samples in May, July, and September 2019 to target the different growth stages of *P. australis* and *S. triqueter* in their wetland ecosystem. We collected 0–10 cm, 10–20 cm, 20–30 cm and 30–40 cm soil samples in each quadrate through the four-point method (one standard sample and three duplicate samples). Meanwhile, we took soil samples for bulk density measurement and put them into the bulk density tank. Labels were affixed on the outside of the tank indicating the sampling location, soil thickness, sampling date, section number, etc. We put the collected soil samples of different depths into marked plastic bags, sealed them and brought them

back to the laboratory. After sorting and removing rotten roots, leaves, and animal remains, the samples were put into a dry and ventilated place to fully air dry.

Soil nutrient determination was conducted according to the method of *Bao (2010)*; TN was determined by the sulfuric acid-accelerator-Kjeldahl method. TP was determined by the sulfuric acid-perchlorate-molybdenum-antimony resistance colorimetric method.

## RESULTS

### Effects of different restored years on seasonal variation and structural characteristics of TN in typical marsh wetland

TN in *P. australis* wetland soil was found to increase gradually through the seasonal effects of the restoration periods, while TN in *S. triqueter* wetland increased first and then decreased during May–September under natural conditions (Fig. 1). Soil TN in *P. australis* wetlands gradually decreased from May to September during the two-year restoration period, while that in *S. triqueter* wetlands gradually increased. Soil TN in *P. australis* wetland decreased first from May to July and then increased sharply from July to September, while the soil TN of *S. triqueter* wetland increased from May to September under the four-year restoration period. The change of soil TN in the *P. australis* wetland was consistent with that of the four year restoration period, while the change of soil total nitrogen in the *S. triqueter* wetland gradually decreased from May to September under the six-year restoration period (Fig. 1).

Soil TN in the *P. australis* wetland decreased as soil depth increasing in both natural conditions and during the restoration period. In May, the soil TN content in the natural *P. australis* and *S. triqueter* wetlands was not high, until the restoration six-year period. The average TN values of 0–10 cm, 10–20 cm, 20–30 cm, and 30–40 cm in *P. australis* and *S. triqueter* wetland soil were 4.0 g/kg, 1.06 g/kg, 0.85 g/kg, and 0.55 g/kg, and 1.35 g/kg, 1.19 g/kg, 1.10 g/kg, and 0.52 g/kg, respectively. In July, the soil TN contents of *P. australis* and *S. triqueter* wetlands were higher than those in May. The *P. australis* soil TN returned to its natural condition during the six-year restoration period. The wetland soil TN at 0–10 cm, 10–20 cm, 20–30 cm, and 30–40 cm was 2.34 g/kg, 059 g/kg, 0.53 g/kg, and 0.44 g/kg, respectively (Fig. 1). The mean of each layer of *S. triqueter* wetland soil TN was consistent with natural conditions at the restored two-year period. TN was 1.08 g/kg, 0.96 g/kg, 0.51 g/kg, and 0.31 g/kg at 0–10 cm, 10–20 cm, 20–30 cm, and 30–40 cm layers, respectively. In September, the TN in each layer of *S. triqueter* wetland soil decreased under natural conditions and during the different restored periods, compared to July. The soil TN of *P. australis* wetlands at 0–10 cm under the natural conditions and in the restored four-year soil increased in September compared to July; these values were 3.11 g/kg and 1.92 g/kg, respectively.

### Effects of different restored years on seasonal variation and structural characteristics of TP in typical marsh wetland

The seasonal effects of restoration years on soil TP of *P. australis* and *S. triqueter* wetlands were investigated. The soil TP in *P. australis* wetlands decreased gradually, while that in the *S. triqueter* wetlands increased first and then decreased under natural condition from May to September. The soil TP of the *P. australis* wetland increased first and then

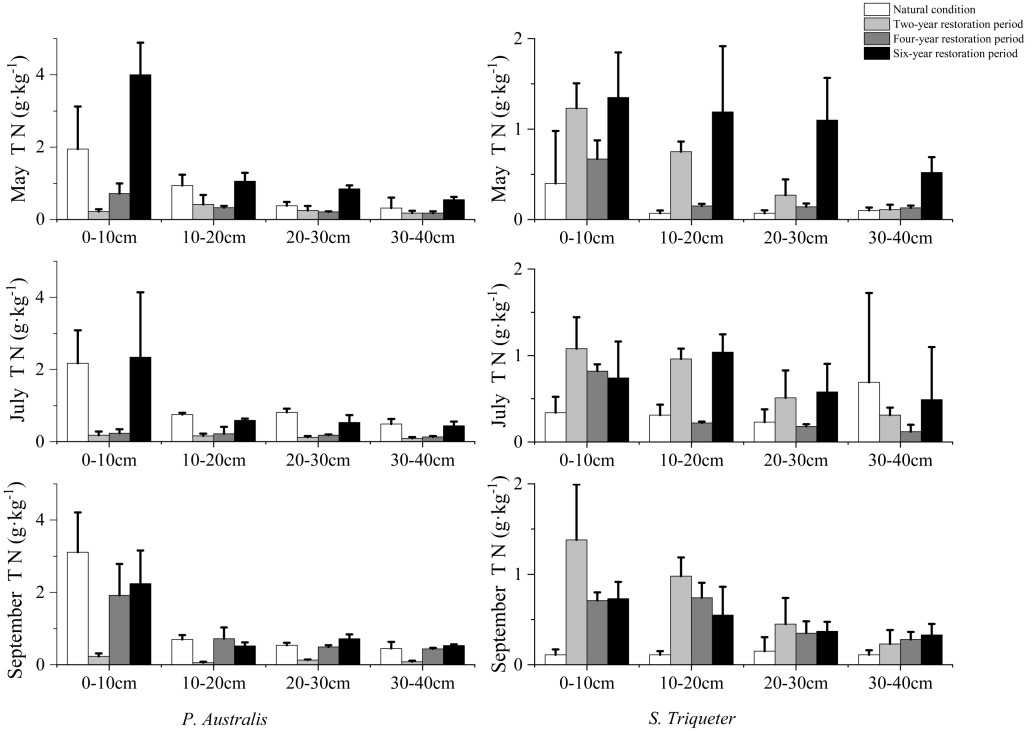

**Figure 1** **Seasonal variation and structural characteristics of TN in typical wetland.**

decreased, while that of the *S. triqueter* wetland increased gradually under the restored two-year period from May to September (Fig. 2). The TP of *P. australis* wetlands decreased first (May–July) and then increased (July–September), while the soil TP of the *S. triqueter* wetlands increased gradually under the restored four year period from May to September. The soil TP of *P. australis* wetlands decreased first (May–July) and then increased (July–September), while that of the *S. triqueter* wetland decreased gradually under the restored six-year period from May to September (Fig. 2).

The soil TP in the *P. australis* wetland decreased as the soil depth increased in the natural condition and restored years. In May, the soil TP in the *P. australis* wetland and *S. triqueter* wetland was still not restored to its natural condition after six years. By July, the *P. australis* and *S. triqueter* wetlands began to reach their natural condition in the two-year restoration period soils, and the average value of wetland soil TP in 0–10 cm, 10–20 cm, 20–30 cm, and 30–40 cm layers were 0.21 g/kg, 0.19 g/kg, 0.28 g/kg, and 0.18 g/kg, and 0.32 g/kg, 0.28 g/kg, 0.21 g/kg, and 0.17 g/kg, respectively (Fig. 2). In September, the soil TP in the *P. australis* wetland displayed a wave of change with increased concentrations under natural conditions that were higher than the restored years (0.33 g/kg, 0.13 g/kg, 0.16 g/kg, and 0.13 g/kg, respectively). The soil TP in the *S. triqueter* wetland under restored conditions was higher than that under natural conditions. For example, the average soil TP in the 0–10 cm, 10–20 cm, 20–30 cm, and 30–40 cm layers in September was 0.36 g/kg, 0.31 g/kg, 0.21 g/kg, and 0.17 g/kg in the two-year restoration, respectively (Fig. 2).

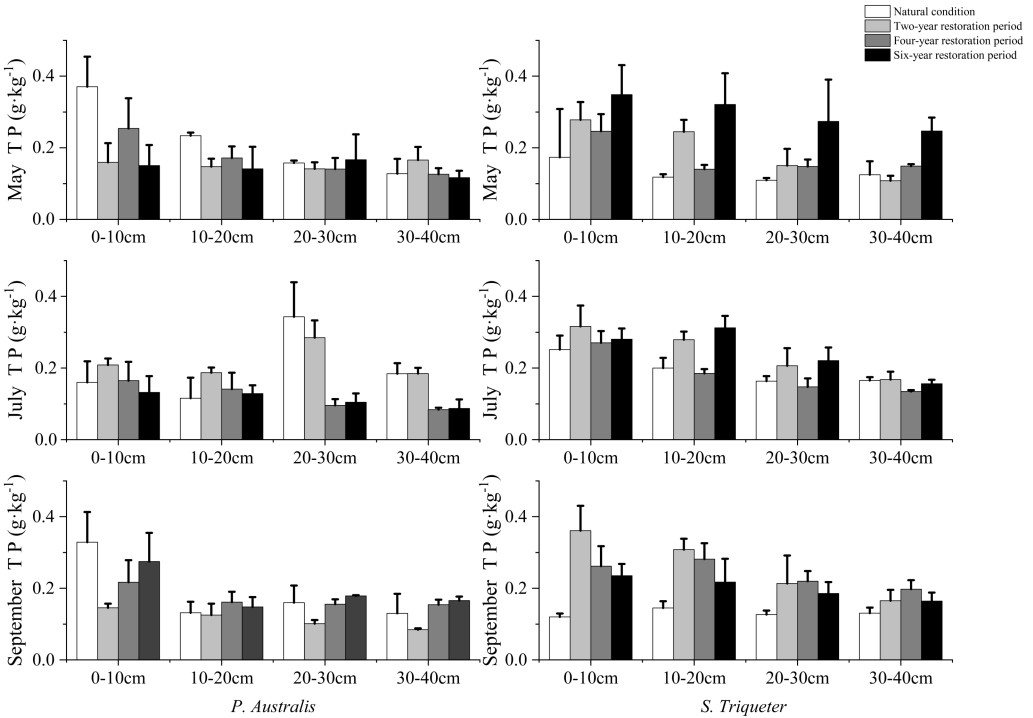

**Figure 2** **Seasonal variation and structural characteristics of TP in typical wetland.**

## Effects of different restoration years on seasonal and structural characteristics of soil N:P in typical wetlands

The seasonal effects of restoration periods on soil N:P in *P. australis* and *S. triqueter* marsh wetlands reveals that under natural conditions, the N:P of *P. australis* wetlands gradually decreased with the increase of soil depth in 0–10 cm, 10–20 cm, and 20–30 cm. It then increased slightly at depths of 30–40 cm from May to September, and the N:P of 0–10 cm was highest at 13.58 in July. The maximum N:P in the *S. triqueter* wetland among different layers was 2.29 (0–10 cm) in May, 4.19 (30–40 cm) in July, and 1.16 (20–30 cm) in September (Fig. 3). In the two-year period, the soil N:P in the *P. australis* wetland increased first and then decreased with the increase of soil depth from May to July. The soil N:P of the *S. triqueter* wetland decreased gradually with increased soil depths from May to September and then reached its natural state (Fig. 3). In the restored four-year period, the soil N:P in the *P. australis* wetland reached its maximum value in July, which exceeded that in the natural condition in July. The soil N:P in the *S. triqueter* wetland decreased with increasing depth from May to September. In the restored six-year period, the N:P in the *P. australis* wetland soil increased first and then decreased with the soil depth from May to July. The N:P in the *S. triqueter* wetland soil decreased first and then increased and then decreased with increasing soil depths in May and September. The N:P of each layer was higher in May (3.87, 3.70, 4.02, and 2.10, in 0–10 cm, 10–20 cm, 20–30 cm, and 30–40 cm, respectively) (Fig. 3).
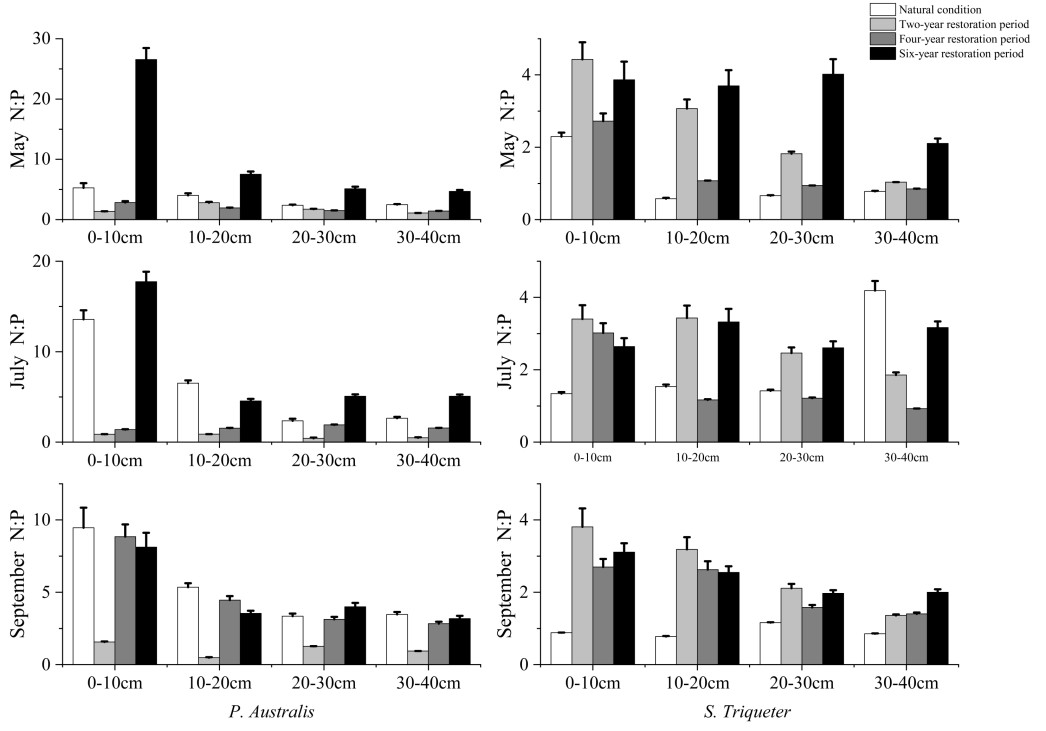

**Figure 3** Seasonal variation and structural characteristics of N:P in typical wetland.

## DISCUSSION

Wetland degradation leads to a decreased capacity to accumulate nutrient elements and, subsequently, poor vegetation growth. After wetland restoration, the accumulation of source elements is gradually restored and the continuous accumulation of C, N and P in *P. australis* was shown to increase with restoration time, indicating the function of the "reservoir" of nutrient elements in wetland restoration (*Zhou, 2021*). Different restoration periods have significantly different effects on the seasonal TN of *P. australis* and *S. triqueter* wetland soil. Our study showed that soil TN in *P. australis* wetlands gradually increased under natural condition from May to September (Fig. 1). These results were consistent with previous studies on the accumulation of soil N in sod saline-alkali *P. australis* wetlands (*Yang et al., 2020*). Our results showed that the N of the *P. australis* wetland soil first decreased from May to July and then dramatically increased from July to September in the four-year restoration period. *Yang et al. (2020)* showed that the soil TN content first decreased and then increased with the growth cycle of plants. The reason may be that soil N in May was derived from the previous year's accumulation because, in this period, the plants were at the beginning growth stage and required less N. With the increase of N required for plant growth, soil N decreased (*Walton et al., 2020*; *Yang et al., 2020*). When the temperature dropped, the *P. australis* began to die and its N demand decreased, which led to soil N accumulation. Part of the N was returned during the decomposition of the litter, and the mineralization of organic N slowed down, leading to the accumulation of

soil nitrogen (*Ye et al., 2016*). The TN content of *P. australis* wetland decreased with the increase of soil depth, which was 0–40 cm in the natural condition and restored state. These results were similar to those of *Yang et al. (2020)*, which showed that the TN content of soil was the highest in the surface layer, and decreased with the increase of soil depth in the 0–100 cm soil layer of the Zhalong wetland in China.

P is an important nutrient element in the wetland ecosystem and its content affects the primary productivity of wetland soils (*Montgomery, Eames & Klimas, 2021*), which are the main source of P nutrients for plants. The P supply can directly affect plant growth and development (*Daneshvar et al., 2017*). The storage capacity of P in restored wetland soil gradually decreases with increased profile depth, and the P storage capacity of the surface soil was significantly higher than that of the underlying soil, which is the main P storage layer (*Song, 2020*). Our results showed that soil TP in *P. australis* wetland first decreased (May–July) and then increased (July-September) in the four-year restoration period (Fig. 2), which was consistent with previous studies. *Bai et al. (2007)* showed consistent results on the seasonal dynamics of soil P in a seaward wetland. Plants grew relatively slowly in May and may have absorbed less P. After June, plants entered a stage of rapid growth and the demand for P increased significantly, leading to a rapid decline in soil TP. Later, the demand for P began to decrease as the plants matured and P accumulation and fixation occurred in the wetland soil (*Jiang & Mitsch, 2020*; *Kang et al., 2021*).

The results showed that the TP in *S. triqueter* wetlands decreased substantially with an increasing soil depth, both in the restored years and under natural conditions (Fig. 2). *Li & Liu (2020)* found that P was mainly enriched in the 0–20 cm soil layer, and the TP content decreased with the increase of soil depth. Additionally, the soil TP content in the restored wetland of *Typha orientalis Presl* gradually increased and approached its natural conditions. The soil TP content was significantly affected by the years of artificial planting of cattails, but was less affected by soil depth, which was consistent with our results (*Li & Liu, 2020*).

Plant tissues adjust their N:P to change their growth rate in order to adapt to environmental changes. The higher N:P in plants reflects the relatively high N and P utilization efficiency of plants, which is an adaptation strategy of wetland plants to the environment with low nutrients (*Niu, Dong & Fu, 2011*). Our results showed that N:P in *S. triqueter* wetland was not affected by restoration years but was significantly affected by soil depth. While the contents of TN and TP gradually decreased with the increase of soil depth, the changes of wetland soil N:P are consistent with the changes of TN in Bayanbulak Alpine Wetland. This study mainly focused on TN and TP during the restored period, and other ecological stoichiometric elements were also important for wetland vegetation restoration and productivity accumulation. In the future, we will further strengthen the research by including all of the biogenic elements of wetland restoration.

## CONCLUSIONS

The restoration period had a significant effect on the soil TN in *P. australis* wetlands. TN gradually increased throughout the growing season under natural conditions, while TN

decreased gradually from May to September during the two-year restoration period. It decreased from May to July and increased drastically from July to September during the four- and six-year restoration period. The soil TP in *S. triqueter* pus wetlands increased during the whole growing season, and during the restoration years, the soil TP of *S. triqueter* wetlands was highest from May to July. In September, the changes of N:P and TN in the *S. triqueter* wetlands maintained the same change trend. With a different soil layer structure, the average soil TN in *P. australis* wetlands was close to its natural conditions for each layer in the six-year restoration period, which were 2.34 g/kg, 0.59 g/kg, 0.53 g/kg, and 0.44 g/kg in 0–10 cm, 10–20 cm, 20–30 cm, and 30–40 cm in July, respectively. The soil TP in the *S. triqueter* wetlands typically decreased with the increase of soil depth and it was higher in the two-year restoration period soils, which were 0.36 g/kg, 0.31 g/kg, 0.21 g/kg, and 0.17 g/kg in 0–10 cm, 10–20 cm, 20–30 cm, and 30–40 cm in September, respectively. The N:P in the *P. australis* wetlands decreased gradually with the increase of soil depth in 0–10 cm, 10–20 cm and 20–30 cm, and then increased slightly in 30–40 cm; the soil N:P at the 0–10 cm depth was up to 13.58 in July.

## ACKNOWLEDGEMENTS

We thank the authors who contributed to this manuscript and the students who participated in the field research test. We also thank the reviewers and journal editors who have contributed to this manuscript.

### Funding

This research was jointly supported by Science and Technology Department Program of Jilin Province of China (YDZJ202201ZYTS474 and 20230508029RC), Jilin Province Foreign Expert Project (L202322) and the National Key R&D Program of China (2022YFF1300900). The funders had no role in study design, data collection and analysis, decision to publish, or preparation of the manuscript.

### Grant Disclosures

The following grant information was disclosed by the authors:
Science and Technology Department Program of Jilin Province of China: YDZJ202201ZYTS474, 20230508029RC.
Jilin Province Foreign Expert Project: L202322.
National Key R&D Program of China: 2022YFF1300900.

### Competing Interests

The authors declare there are no competing interests.

### Author Contributions

- Dandan Zhao conceived and designed the experiments, performed the experiments, analyzed the data, prepared figures and/or tables, and approved the final draft.

- Daiji Wan performed the experiments, prepared figures and/or tables, and approved the final draft.
- Jian Yang conceived and designed the experiments, analyzed the data, prepared figures and/or tables, and approved the final draft.
- Jiping Liu conceived and designed the experiments, analyzed the data, authored or reviewed drafts of the article, and approved the final draft.
- Zhicheng Yong conceived and designed the experiments, authored or reviewed drafts of the article, and approved the final draft.
- Chongya Ma analyzed the data, authored or reviewed drafts of the article, and approved the final draft.

## Data Availability

The raw measurements are available in the Supplementary File.

## Supplemental Information

Supplemental information for this article can be found online at http://dx.doi.org/10.7717/peerj.16766#supplemental-information.

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
