# Peer review of "Effects of restoration years on soil nitrogen and phosphorus in inland salt marshes"

_PeerJ, doi:10.7717/peerj.16766_

## Round 0.1 · original submission · Major Revisions

Your manuscript has been reviewed by two experts in this research field. Both reviewers found interesting results and thought your manuscript was suitable for our journal. However, both of them do have a few comments and suggestions. Please address these comments carefully and prepare a point-to-point response. I look forward to receiving your revision.

**Language Note:** The review process has identified that the English language must be improved. PeerJ can provide language editing services - please contact us at copyediting@peerj.com for pricing (be sure to provide your manuscript number and title). Alternatively, you should make your own arrangements to improve the language quality and provide details in your response letter. – PeerJ Staff

·

Basic reporting

The ecological fragility of inland salt marsh wetlands and the serious salinization and desertification lead to the continuous decline of wetland productivity and the decrease of biodiversity, showing a serious degradation trend. Wetland soil nutrient content is an important factor to determine soil fertility, and its dynamic change significantly affects the productivity of wetland ecosystem. The manuscript of dynamic changes of soil nutrients in inland salt marshes at different restoration stages provides theoretical guidance and reference for ecological restoration of degraded wetlands and sustainable management and protection of wetlands in this area. However, this manuscript still needs to be further improved and discussed for minor revision, as follows:

1.Some English sentences should be further polished and modified by the authors.
2.“Effects of restoration years on soil nitrogen and phosphorus in inland salt marshes” Does this title need to take into account the seasonal and structural characteristics of N and P?
3.Introduction should be appropriately presented in the summary.
4.Recent or more advanced research in the field of wetland restoration should be added to the introduction and strengthened the logic of the introduction .
5.Specific information such as the location of the study area can be indicated by appropriate maps.
6.The results need to be further condensed to summarize the most significant regularities.
7.The paper discusses large sectors that are combined with regional relevant research, for example, "soil TN in Phragmites australis wetland gradually increased in the natural condition during May-September." Whether it can be combined with other regional or large-scale related studies for further discussion.
8.The citation format of the references in the text is inconsistent, and the manuscript needs to be standardized and strengthened. For example, line 112.

Experimental design

no comment

Validity of the findings

no comment

Reviewer 2 ·

Basic reporting

Zhao et al. assessed the effects of restoration years on soil nitrogen and phosphorus in inland salt marshes of Northeast China. This topic is interesting and the field work is very gruelling, which deserve encouraging and revision opportunities.

Experimental design

The design needs further clarification. Please check the general comments below.

Validity of the findings

The findings were acceptable with the support of field monitoring data.

Additional comments

-L22: The botanic name of Scirpus triqueter L. is a synonym of the species: Schoenoplectus triqueter (L.) Palla. Please check which one is more formal. Italic is not required for L., and Scirpus should be short for S. after the first occurrence. The same for Phragmites australis.
-L16: How to examined the seasonal and profile effects of different restoration years?
-L19-27: The key data should be added properly.
-L108-114: Should be more concise for the descriptions of study area, for these sentences should be moved to the section below.
-L148: The restoration methods should be added briefly.
-L155: What’s the patent name of (ZL201210534424.7)?
-L165-179: the references of TN and TP test methods should be added.
-L181: No need to repeat each data copied from the figures.
-L260: The figure number(s) should be cited in Discussion section to support your ideas.
-L319: Respond the three hypotheses in Introduction directly.
-Figures: Too many. Figures 1-3 can be merged in one figure with 2*3 subgraphs. The asterisks for significant differences were unclear. Why the result descriptions occurred after each figure caption?

---

## Round 0.2 · accepted · Accept

Please pay attention to Reviewer's minor comments and modify your manuscript accordingly while you prepare the final version, such as the legend “restored two years”, i.e., "two years after restoration", or "restoring for two years"?, as well as significant differences.

Reviewer 2 ·

Basic reporting

I’m pleased to see the substantial changes by the authors. I recommend it to be accepted after minor revision. Only a small question for the figures.
Figures:
-The legend “restored two years”: Two years after restoration, or restoring for two years?
-Significant differences: Much clearer, whereas why no letters or asterisks above the error bars?I recommend it to be accepted after minor revision.

Experimental design

The design has been clarified.

Validity of the findings

The findings were acceptable.

Additional comments

No comment.